# Causal-driven Large Language Models with Faithful Reasoning for Knowledge Question Answering

## ABSTRACT

In Large Language Models (LLMs), text generation that involves knowledge representation is often fraught with the risk of "hallucinations", where models confidently produce erroneous or fabricated content. These inaccuracies often stem from intrinsic biases in the pre-training stage or from the incorporation of human preference biases during the fine-tuning process. To mitigate these issues, we take inspiration from Goldman's causal theory of knowledge, which asserts that knowledge is not merely about having a true belief but also involves a causal connection between the belief and the truth of the proposition. We instantiate this theory within the context of Knowledge Question Answering (KQA) by constructing a causal graph that delineates the pathways between the candidate knowledge and belief. Through the application of the do-calculus rules from structural causal models, we devise an unbiased estimation framework based on this causal graph, thereby establishing a methodology for knowledge modeling grounded in causal inference. The resulting CORE framework (short for "Causal knOwledge REasoning") is comprised of four essential components: question answering, causal reasoning, belief scoring, and refinement. Together, they synergistically improve the KQA system by fostering faithful reasoning and introspection. Extensive experiments are conducted on ScienceQA and HotpotQA datasets, which demonstrate the effectiveness and rationality of the CORE framework.

## CCS CONCEPTS

• **Computing methodologies** → **Computer vision representations**; **Natural language generation**; **Reasoning about belief and knowledge**.

## KEYWORDS

Large Language Models, Causal Theory of Knowledge, Knowledge Question Answering

## 1 INTRODUCTION

With the development of instruction tuning techniques, large language models (LLMs) have demonstrated impressive performance across tasks in natural language understanding and generation. These models have shown great potential in human-like capabilities of common sense, logical reasoning and tool manipulation, which can understand and harness knowledge to improve problem-solving

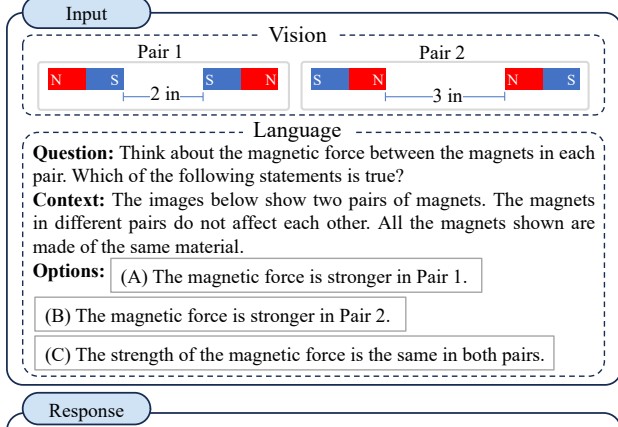

**Figure 1: A typical example of a KQA task, which requires in-depth knowledge of multimodal information to answer the question.**

and decision-making tasks. However, many studies [3, 5, 8, 11, 18] have found that LLMs may produce some hallucinations that are neither accurate nor factual. Hallucinations in LLMs occur when they produce incorrect or misleading information, often linked to shortcomings in two types of knowledge: intrinsic and extrinsic. Intrinsic knowledge refers to common sense, and logical reasoning abilities, which LLMs may sometimes fail to utilize, leading to nonsensical or factually wrong outputs. Extrinsic knowledge is about grasping new, contextual, or specific tool-based information, which LLMs may not always interpret accurately. Influenced by biases in their training data, both knowledge types can lead to LLMs delivering coherent, yet factually incorrect or misleading responses.

To address these issues, there are mainly two types of research works proposed to reinforce LLMs, i.e. reasoning-based method and confidence-based method. Reasoning-based methods [6, 12, 28, 32, 40, 45] are proposed to enhance the capability of LLM not by retraining models but through reasoning with knowledge of LLMs, allowing LLMs to improve decision-making on a continual thought process. Confidence-based methods [14, 17, 21, 23, 31, 34, 35] are designed to explore various reasoning paths and employ a verifier that checks the correctness or self-consistency of each reasoning step to leading a unique answer throughout the multi-path reasoning. While both types of methods have been found to be effective, there is a notable absence of a rational explanation regarding the necessity of incorporating all these designs. In addition, whether these two approaches are intrinsically related still needs to be further explored.

In fact, there is a specific philosophical definition of "knowledge", which has been discussed comprehensively in Goldman's causal theory of knowledge [10]. The causal theory posits that true knowledge is derived from a causal connection between a proposition and a belief. In the LLM context, the "Proposition" represents the input and output information from LLMs. The "Causal Connection" is the critical step, wherein the LLM must establish a logical and factual link between the proposition and the potential belief. This involves not only recalling information from its vast training dataset but also applying logical reasoning to ensure that the connections made are rational and grounded in causality. Finally, the "Belief" is the confidence to the proposition of whether it is a faithful knowledge, which should be a true reflection of the causal and logical processing undertaken in the previous step.

Drawing from Goldman's causal theory of knowledge, we propose a CORE framework (short for "Causal knOwledge REasoning") to process Knowledge Question Answering (KQA) tasks, in which a typical KQA process is as illustrated in Figure 1. The CORE framework suggests a reasoning structure where each decision or output follows a clear path: Proposition → Causal Connection → Belief. To this end, we translate the reasoning structure into a practical application by creating a causal graph that outlines the connections among knowledge theory and training biases of LLMs. Meanwhile, we apply do-calculus rules from structural causal models [27] to derive an unbiased estimation framework for this causal graph. Based on the estimation formulation, the CORE framework consists of four key components, each playing a crucial role in enhancing KQA systems with robust and reliable reasoning capabilities: 1) Question Answering. This module aims to provide a preliminary answer hypotheses to a KQA question as illustrated in Figure 1. 2) Causal Reasoning. This component focuses on establishing a logical and factual link between a given question-answering response (the Proposition) and the potential confidence (the Belief). 3) Belief Scoring. This module assesses the logical consistency and faithful accuracy of KQA output, aligning with the principles of causal reasoning. 4) Refinement. This last component fine-tunes the outputs, ensuring that the reasoning process aligns with the causal mechanisms outlined in the previous causal reasoning process.

By incorporating the CORE framework, LLMs can be guided to process information in a manner that mimics human-like causal reasoning to produce knowledge, termed as "Faithful Reasoning" [6, 12, 13], potentially reducing instances of hallucinations and enhancing the model's ability to generate more accurate and reliable responses. This approach aligns with the intrinsic and extrinsic knowledge frameworks, ensuring that the LLM's reasoning is not only based on the data it has been trained on but also the logical coherence and factual accuracy of the information. Further, this "Faithful Reasoning" method can synergize with existing reasoning-based and confidence-based methods. By integrating causal theory into these approaches, LLMs can be better equipped to navigate the complex reasoning paths, assess the self-consistency of their outputs, and arrive at conclusions that are not only confident but also causally and logically sound. By conducting experiments on two real-world datasets, we have illustrated the superiority of our proposed framework on both overall performance comparison and micro-scope studies.

The main contributions of this work are summarized as follows:

- We introduce a causal knowledge reasoning framework, which is a novel approach integrating Goldman's causal theory of knowledge into large language models, particularly in the KQA context. This framework is designed to enhance the reliability and accuracy of knowledge understanding in LLMs by establishing a clear pathway from proposition to belief through causal connections.
- Causal graph and do-calculus rules from structural causal models are employed to create an unbiased estimation framework within LLMs. Four key components are constructed, namely, Question Answering, Causal Reasoning, Belief Scoring, and Refinement, which allow for a more reliable and accurate knowledge representation and understanding in LLMs, leading to more faithful reasoning capabilities.
- Extensive experiments are conducted on ScienceQA and HotpotQA datasets, which demonstrate the rationality and effectiveness of the CORE method. Additionally, we will make the implementations available to the research community to facilitate further research[1].

## 2 RELATED WORKS

### 2.1 Knowledge in LLMs

The exploration of knowledge in pre-trained LLMs has been a central theme in recent research, reflecting the advances and challenges in harnessing LLMs for enhancing decision-making and problem-solving in multimodal contexts.

Yang et al. [38] examine the capabilities of GPT-4V, a model that augments LLMs with multi-sensory skills, including visual understanding. This research demonstrates the potential of LLMs in harnessing both intrinsic and extrinsic knowledge, highlighting their utility in diverse applications. Similarly, Yin et al. [42] provide an extensive survey on multimodal large language models, elucidating their surprising emergent capabilities and potential paths toward artificial general intelligence. These studies collectively showcase the significant strides made in LLMs, especially in terms of multimodal understanding and logical reasoning. On the other hand, Davis and Aaronson [8] bring to light the limitations of LLMs in interfacing with external tools like Wolfram Alpha, particularly in solving math and science problems. Their findings resonate with the challenges LLMs face in processing intrinsic and extrinsic knowledge accurately, often leading to incorrect or misleading outputs. Complementing this perspective, Arkoudas [3] critically evaluates GPT-4's reasoning abilities, underlining its shortcomings in logical reasoning and problem-solving, which are core aspects of intrinsic knowledge.

These studies collectively underscore the dual nature of LLMs' advancements. On one hand, they demonstrate significant progress in multimodal understanding and logical reasoning. On the other hand, they reveal the persistent challenges of hallucinations and biases, emphasizing the need for improved models that can accurately harness intrinsic and extrinsic knowledge without succumbing to inaccuracies and misleading information. To this end, we aim to enhance the reasoning capabilities of LLMs, ensuring more reliable and accurate knowledge processing in line with the principles of "Faithful Reasoning".

---

[1]Please see the supplementary files in the ACMMM submission system.

## 2.2 Faithful Reasoning in LLMs

Advancements in LLMs have been centered around enhancing "Faithful Reasoning", mainly categorized into reasoning-based and confidence-based methods.

**Reasoning-Based Methods.** This category includes methods that improve intrinsic reasoning capabilities of LLMs. Chain-of-thought prompting [32], least-to-most prompting [45], and the rethinking with retrieval approach [12] exemplify this. These methods break down complex problems into simpler steps, enhancing logical processing abilities of LLMs. The tree of thoughts framework [40] and Reflexion [28] also fall under this category, promoting adaptive learning and strategic decision-making in LLMs.

**Confidence-Based Methods.** Focused on verifying the reasoning processes, this category includes DIVERSE [23], discriminator-guided reasoning [14], and self-evaluation guided beam search [35]. These methods ensure the reliability and accuracy of LLM outputs by verifying each reasoning step and employing feedback mechanisms. Self-consistency methods [1, 31, 34] further exemplify this approach by marginalizing out multiple reasoning paths to select the most consistent answer.

In light of these developments, we provide a causal-based perspective grounded in Goldman's causal theory [10] and Pearl's structural causal model [27], elucidating why these faithful reasoning methods are effective. Based on a theoretical unbiased formula, we propose a CORE framework with four main components, which synergistically combines these two methodological approaches to enhance the reliability and interpretability of LLMs in processing and generating knowledge.

## 2.3 Knowledge Question Answering

The KQA task, particularly in its multimodal form, presents a unique challenge and opportunity for language modeling systems. It combines objective knowledge extraction with reasoning across multiple modalities, making it an ideal testbed for advanced multimodal LLMs.

Current studies [2, 7, 9, 16, 19, 20, 24, 43, 44] mainly focus on enhancing multimodal reasoning capabilities in KQA task. Innovations such as BLIP models [7, 19, 20], ScienceQA [24], and Multimodal-CoT [44] exemplify this trend, showcasing significant advancements in the LLM ability to process and reason within text-rich visual contexts and using multimodal information for more accurate and comprehensive question-answering.

In contrast to these methods, our causal-driven approach in the CORE framework potentially offers a unified solution for VQA tasks. By grounding LLM reasoning in causal relationships as per Goldman's causal theory, our approach aims to validate and generate faithful knowledge, which seamlessly integrate and reason across diverse knowledge forms, from text to visual content, in a causally faithful manner.

## 3 METHODS

This section describes the formulation of an interventional causal model for the KQA task within a visual-question answering (VQA) context, grounded in Goldman's causal theory [10] and Pearl's structural causal model [27]. Based on this theoretical method, we build a CORE framework to facilitate faithful reasoning by

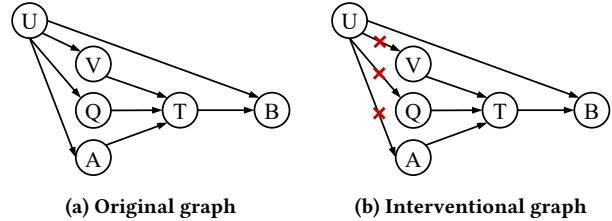

(a) Original graph          (b) Interventional graph

**Figure 2: The proposed interventional causal graph based on Goldman's causal theory of knolwedge and Pearl's structural causal model. Causal graph: V (Visual), Q (Question), A (Answer), T (Causal Thoughts), B (Belief), and U (Unobservable Confounder i.e. Dataset Bias)**

implicitly accounting for confounders without direct modeling, which mitigates dataset biases and foster the acquisition of faithful knowledge.

## 3.1 Causal Model of Knowledge in LLMs

*3.1.1 Causal Graph.* As Goldman's causal theory of knowledge posits, true knowledge is derived from a causal connection between a proposition and a belief. Since the knowledge in LLMs is acquired during the pre-training stage, this may introduce dataset bias in the models, leading to inaccurate knowledge understanding. To confront these problems, we employ Pearl's structural causal model to obtain a causal graph between the causal theory and the learning process of LLMs, which is illustrated in Figure 2a, where nodes and edges represent endogenous variables and causal relationships, respectively. To justify the rationality of the proposed causal graph, we give detailed interpretations of the causal graph as follows.

$\{V, Q, A\} \rightarrow T \rightarrow B$. The sequence of nodes $V, Q, A$ represents the input variables: the Visual information of images (V), the Textual Question (Q), and the Answer (A), in which these variables combine to form a VQA example. Causal Thoughts (T) are the reasoning process about how the answer causally relates to the question within the given multimodal context. The flow from T to Belief (B) represents the evaluation of the causal thoughts, leading to a belief about the accuracy of answer. This progression from question and visual context, through causal reasoning, to belief is central to our CORE approach to understanding and reasoning knowledge.

$U \rightarrow \{V, Q, A, B\}$. The Unobservable Confounder (U), often representative of Dataset Bias, is an external factor that can influence the belief in the truth of a VQA proposition as knowledge. It is important to recognize the potential of U to introduce training bias into the understanding of knowledge, which can lead to incorrect beliefs. Our CORE framework aims to address this by isolating and mitigating the influence of U on the causal pathway from inputs to belief, thus striving for a more accurate and unbiased reasoning process.

*3.1.2 Causal Intervention and Estimation.* Given images and questions, LLM-based VQA models can perform natural language generation to answer the questions. However, the generated answers may be incorrect because of the training bias of the LLMs. To further verify the beliefs of the generated knowledge based on the causal model, we aim to estimate the true belief of knowledge using LLMs

without the impact of the training biases. Therefore, we need to estimate the interventional conditional probability of the beliefs of whether the answers correctly answer the questions, which is not equal to the observational conditional probability:

$$P(b|do(v), do(q), do(a)) \neq P(b|v, q, a), \quad (1)$$

where the $do(\cdot)$ operation for input variables show that we want to pursue the true causal effect of knowledge $\{V, Q, A\}$ to beliefs $B$ by intervention [27]. This circumvents hallucinations from biases derived from $U$ and ensures the reliability of the model outputs within the KQA tasks.

To this end, we employ causal intervention to disentangle the influence of dataset bias (i.e., the unobservable confounder $U$) from the knowledge reasoning process, as depicted in Figure 2b. We adopt do-calculus rules [27] to derive an unbiased belief estimation, focusing on a causal-driven process that considers only the variables of interest while controlling the confounding influences. Let $\hat{\cdot}$ be the intervention operator $do(\cdot)$ for blocking the dataset bias, the causal intervention process can be described mathematically as follows:

$$
\begin{aligned}
& P(b|\hat{v}, \hat{q}, \hat{a}) \\
=& \sum_t P(b|\hat{v}, \hat{q}, \hat{a}, t) P(t|\hat{v}, \hat{q}, \hat{a}) \\
\overset{rule2}{=}& \sum_t P(b|\hat{v}, \hat{q}, \hat{a}, \hat{t}) P(t|\hat{v}, \hat{q}, \hat{a}) \\
\overset{rule2}{=}& \sum_t P(b|\hat{v}, \hat{q}, \hat{a}, \hat{t}) P(t|v, q, a) \\
\overset{rule3}{=}& \sum_t P(b|\hat{t}) P(t|v, q, a) \\
=& \sum_t \sum_{v',q',a'} P(b|\hat{t}, v', q', a') P(v', q', a'|\hat{t}) P(t|v, q, a) \\
\overset{rule2}{=}& \sum_t \sum_{v',q',a'} P(b|t, v', q', a') P(v', q', a'|\hat{t}) P(t|v, q, a) \\
\overset{rule3}{=}& \sum_t \sum_{v',q',a'} P(b|t, v', q', a') P(v', q', a') P(t|v, q, a). \quad (2)
\end{aligned}
$$

This derivation process is similar to the methods [30, 33, 37] based on the front-door criterion [27], which transforms the intervention probability estimation task with the $do(\cdot)$ operator into an association probability estimation task. Since $P(t|v, q, a)$ is an encoder for mapping multimodal question-answering inputs to causal thoughts, then the probability of $P(t_{v,q,a}|v, q, a)$ equals to one if this causal reasoning model is assumed to have an deterministic thought process $t_{v,q,a}$ to explain how well the answer response to the question. This means that Equation 2 can be further transformed as follows:

$$
\begin{aligned}
& P(b|\hat{v}, \hat{q}, \hat{a}) \\
=& \sum_t \sum_{v',q',a'} P(b|t, v', q', a') P(v', q', a') P(t|v, q, a) \\
=& P(t_{v,q,a}|v, q, a) \underset{v',q',a'}{\mathbb{E}} P(b|t_{v,q,a}, v', q', a'). \quad (3)
\end{aligned}
$$

This formula indicates that the probability of the true belief $b$ of a proposition $\{v, q, a\}$ can be unbiasedly calculated by averaging the expectations over supporting samples that follow the same causal thought process $t_{v,q,a}$. Therefore, identifying different proposition

$\{v', q', a'\}$ that support the causal explanation $t_{v,q,a}$ is the key to the estimation of the true belief. In practical question-answering scenarios, the visual $v'$ and question $q'$ generally remain constant within a given context, meaning $v' = v$ and $q' = q$. This implies that our estimation should primarily focus on exploring diverse answers $a'$ that coincide with the established causal thought process $t_{v,q,a}$. This recognition plays an important role in estimating the confidence in knowledge generated by the model in response to a visual-question inquiry.

## 3.2 CORE Framework

Based on the unbiased estimation of the interventional target of knowledge as outlined in Equation 3, we aim to estimate it within the context of the VQA task. Given textual inquiry $q$ and the corresponding visual information $v$, we first generate an answer $a$ to the question, and then propose the causal thought process $t$ to support or repel the proposition $\{v, q, a\}$. After that, our goal is to refine the same question $\{v, q\}$ using different perspective responses $a'$ to eventually improve the expected accuracy of beliefs $b$.

To operationalize this theory in VQA, we develop the CORE framework, which integrates four principal components: Question Answering $P(a|v, q)$, Causal Reasoning $P(t|v, q, a)$, Belief Scoring $P(b|v, q, a, t)$, and Refinement $P(a'|v, q, a, t)$. As illustrated in Figure 3, each component serves a unique function in the faithful reasoning process, working collaboratively to ensure a reliable estimation of the interventional target. Below, we detail the specific functions of each component and outline their implementation within the prompting strategy.

*3.2.1 Question Answering.* The Question Answering $P(a|v, q)$ module is the foundational component of our framework, tasked with generating preliminary answer hypotheses based on visual inputs $v$ and questions $q$. Leveraging large language models, this module synthesizes information across modalities to produce contextually relevant answer.

Similar to the previous works [24], we first employ the state-of-the-art visual-language model InstructBLIP [7] to represent the visual context of image as text, where the model has been instruction-tuned on a wide variety of tasks based on the pretrained BLIP-2 model [19, 20]. And then, the visual context is concatenated to the question context to form a visual-question input. To provide a preliminary answer hypotheses, we follow the approach of ScienceQA [24], which perform multi-option question answering task based on a few-shot learning prompt technique. Specifically, we build instructions using two in-context examples with components of the question text, options, and the correct answer text. This style of prompt enables the LLMs to answer specific questions without parameter updates.

*3.2.2 Causal Reasoning.* To verify the faithful accuracy of the answer, our CORE framework emphasizes establishing a causal connection between a given question-answering response and the corresponding belief. Central to our CORE framework, the Causal Reasoning $P(t|v, q, a)$ constructs causal thoughts $t$ that serve as explanatory narratives that connects questions and visual cues to potential answers.

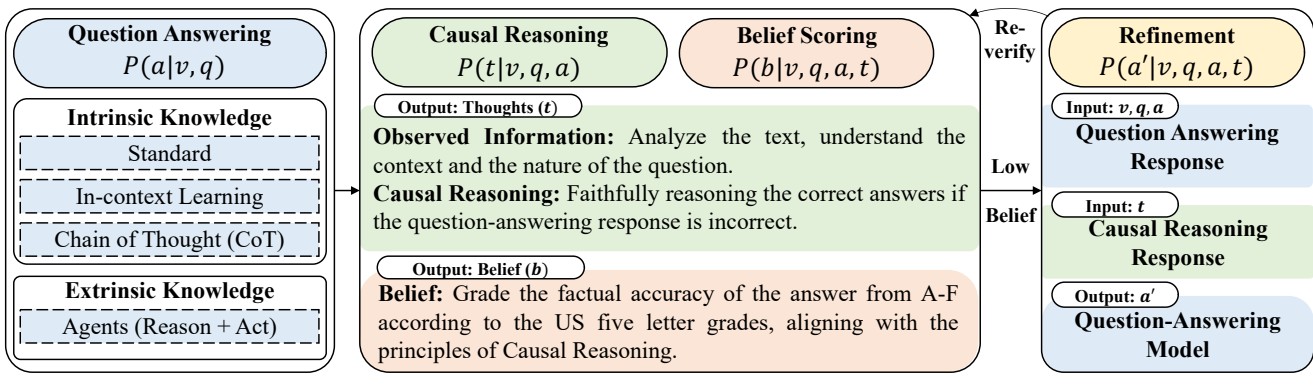

**Figure 3: Comprehensive schematic of the CORE framework, which details the sequential interaction of the four primary components.[2]**

In this work, we embodies the essence of causal reasoning processes as illustrated in Figure 3, enabling the model to observe information and discern logical pathways by prompting the LLMs. By integrating chain-of-thought prompting techniques to facilitate causal reasoning, our Causal Reasoning module perform introspection and evaluate answer hypotheses through a causally-aware process.

*3.2.3 Belief Scoring.* The Belief Scoring module $P(b|v, q, a, t)$ evaluates the faithful accuracy and logical consistency of the generated answers. It assigns belief scores $b$ to the proposition and the causal thought process, reflecting the degree of confidence in the correctness and relevance of candidate knowledge.

To estimate the belief score effectively, our approach employs a verbalized method for estimating model confidence. This method is inspired by recent observations [29, 36] that LLMs have been shown to offer more accurate measure of confidence using output tokens, typically better-calibrated than methods using traditional conditional probabilities. As shown in the prompt text in Figure 3, we adopt a grading scale akin to the US five letter grades to quantify the confidence of the initial answer being correct. Specifically, "A" indicates Confidently Correct 100%, "B" indicates Correct with Moderate Confidence [75%, 1), "C" indicates Incorrect with Some Confidence [50%, 75%), "D" indicates Confidently Incorrect. [25%, 50%), and "F" indicates No Direct Answer [0%, 25%).

*3.2.4 Refinement.* As the final stage of our reasoning process, the Refinement $P(a'|v, q, a, t)$ will polish the initial response. It harmonizes initial answer $a$ with the causal thought $t$ to generate a new answer $a'$ that conform the most plausible causal explanations.

To implement this, we repurpose the prompting method of Question Answering module. By appending the original VQA response and associated causal thoughts to the prompt of the Refinement model, we guide the LLMs to correct their answers according to the established reasoning path.

*3.2.5 Causal-driven Faithful Reasoning.* The CORE framework operates through four main parts, as outlined in Algorithm 1. It starts with the Question Answering module, which generates an initial

---

**Algorithm 1** The pseudo code of CORE framework

**Input:** Textual inquiry $q$, Visual information $v$
**Output:** Refined answer $a_{best}$
    **Question Answering** $P(a|v, q)$
1: Generate initial answer $a$ to the question $q$ given the visual context $v$.
    **Causal Reasoning** $P(t|v, a)$**, Belief Scoring** $P(b|v, q, a, t)$
2: Propose causal thought process $t$ to support or challenge the proposition $\{v, q, a\}$.
3: Assign a belief score $b$ to the initial answer $a$ based on the causal thought process $t$.
    **Refinement** $P(a'|v, q, a, t)$
4: **if** Belief score $b \geq$ Threshold $\tau$ **then**
5:     Refine the answer $a$ to $a_r$ based on the causal thought $t$.
6:     Re-evaluate $a_r$ using causal reasoning and belief scoring module to get belief score $b_r$.
7:     **if** $b_r \geq b$ **then**
8:         $a_{best} = a'$
9:     **else**
10:         $a_{best} = a$
11:     **end if**
12: **else**
13:     $a_{best} = a$.
14: **end if**
    **Return** Best Answer $a_{best}$.

---

answer. Following this, the Causal Reasoning module analyzes the answer to identify the underlying causal rationale. After that, the Belief Scoring module evaluates the confidence level of the model in the initial answer, based on how accurate and consistent it aligns with the causal explanations. If any answers lack sufficient confidence, the Refinement module helps by adjust them to make more sense based on the initial answer and the causal explanations.

Note that we can perform refinement process when the belief score of the answers is lower than a pre-defined belief threshold $\tau$, so as to make a trade off between accuracy and efficiency. And we set $\tau = A$ for better accuracy by default. In addition, when an answer undergoes refinement, the refined answer is not simply

---

[2]Note that the figure mainly represents the core structure of the prompt text. For comprehensive details and specific prompts, please refer to the appendix.

| Model | NAT | SOC | LAN | TXT | IMG | NO | G1-6 | G7-12 | Avg |
|---|---|---|---|---|---|---|---|---|---|
| Human | 90.23 | 84.97 | 87.48 | 89.60 | 87.50 | 88.10 | 91.59 | 82.42 | 88.40 |
| ViLT [16] | 60.48 | 63.89 | 60.27 | 63.20 | 61.38 | 57.00 | 60.72 | 61.90 | 61.14 |
| Patch-TRM [25] | 65.19 | 46.79 | 65.55 | 66.96 | 55.28 | 64.95 | 58.04 | 67.50 | 61.42 |
| VisualBERT [22] | 59.33 | 69.18 | 61.18 | 62.71 | 62.17 | 58.54 | 62.96 | 59.92 | 61.87 |
| UnifiedQABase [15] | 68.16 | 69.18 | 74.91 | 63.78 | 61.38 | 77.84 | 72.98 | 65.00 | 70.12 |
| UnifiedQABase w/ CoT [24] | 71.00 | 76.04 | 78.91 | 66.42 | 66.53 | 81.81 | 77.06 | 68.82 | 74.11 |
| GPT-3.5-A | 74.69 | 74.13 | 79.73 | 74.88 | 71.59 | 78.26 | 78.45 | 71.26 | 75.88 |
| GPT-3.5-AE | 77.49 | 73.79 | 81.82 | 77.17 | 72.29 | 81.18 | 81.46 | 71.32 | 77.84 |
| GPT-3.5-ALE | 77.44 | 74.02 | 83.09 | 77.32 | 72.38 | 81.88 | 81.17 | 72.84 | 78.19 |
| CORE-A | 78.37 | **76.04** | 81.73 | **77.86** | **72.98** | 82.23 | 81.53 | **73.76** | 78.76 |
| CORE-AE | 77.98 | 74.47 | 81.45 | 77.03 | 71.54 | 82.30 | 81.39 | 72.31 | 78.14 |
| CORE-ALE | **78.60** | 75.82 | **82.91** | 77.71 | 72.63 | **83.48** | **82.27** | 73.50 | **79.13** |

Table 1: Performance evaluation of various models across different classes in terms of accuracy (%) on ScienceQA dataset. Format names: A = answer, AE = answer with explanation, ALE = answer with lecture and explanation. Question classes: NAT = natural science, SOC = social science, LAN = language science, TXT = text context, IMG = image context, NO = no context, G1-6 = grades 1-6, G7-12 = grades 7-12. Segment 1: Human performance; Segment 2: VQA baselines; Segment 3: UnifiedQA baselines; Segment 4: GPT-3.5 baselines; Segment 5: Our CORE results. Results in bold are the best performance.

accepted; it undergoes reevaluation through the causal reasoning and belief scoring processes, consistent with the formulation $P(b|t_{v,q,a}, v', q', a')$ established in Equation 3. The revised answer, denoted as $a'$, is adopted only if it achieves a belief score superior to its initial counterpart. Otherwise, the initial response is retained. This iterative refinement process ensures that the framework outputs are both consistent and deeply rooted in a robust causal understanding, thus enhancing the overall reliability of the knowledge they represents.

Overall, all these processes are inspired by the unbiased knowledge evaluation process as shown in Equation 3. For more examples and details on how each part works, please check out the case study (section 4.6) and appendix.

## 4 EXPERIMENTS

### 4.1 Experimental Settings

*4.1.1 Datasets.* Our methodology is evaluated on two prominent datasets, each with its unique structure and challenges. The first, ScienceQA [24], is a comprehensive multimodal science question dataset with annotations that include detailed lectures and explanations. It encompasses a broad domain diversity with 21k multiple-choice questions spanning 3 subjects, 26 topics, 127 categories, and 379 skills. The dataset is organized into distinct training, validation, and test sets containing 12,726, 4,241, and 4,241 examples, respectively.

The second dataset, HotpotQA [39], consists of 113k Wikipedia-based question-answer pairs. Its defining characteristics are as follows: (1) it necessitates the retrieval and synthesis of information across multiple documents to construct answers, and (2) the questions are varied, avoiding restrictions from any predefined knowledge bases or schemas. This dataset provides an additional level of complexity to our evaluation, allowing us to test more advanced prompting techniques on our proposed method. In the following experiments, we randomly select a subset of 1,000 samples for performance evaluation.

*4.1.2 Implementation Details and Baselines.* For the ScienceQA dataset, we establish two-shot in-context learning baselines, following the format of "A", "AE", and "ALE". This involves concatenating question text, context of text and visual, and multiple options as input, with predictions for answers (A), lectures (L), and explanations (E) as output, aligning with the previous works [24, 44]. These prompting formats are employed in our Question Answering module and the output formatting of the Refinement module for a fair comparison. Following the ScienceQA settings [24], our baselines include VQA models such as ViLT [16], Patch-TRM [25], and VisualBERT [22]; Text-to-text Language Models (LMs) like UnifiedQA [15]; and our multimodal models based on GPT-3.5 [4, 26] and InstructBLIP [7].

In evaluating our CORE framework on the HotpotQA dataset, we further introduce three baselines using different prompting techniques: standard prompt, zero-shot chain-of-thought [32], and ReAct [41], with the latter having the capability to access Wikipedia information through tool manipulation. This allows us to perform a comparative analysis of different reasoning models with and without the application of our Faithful Reasoning (FR) components (e.g. Causal Reasoning, Belief Scoring, and Refinement). The same prompting formats are utilized correspondingly in both our Question Answering module and for output formatting in the Refinement module, ensuring the fairness in comparative analysis. Note that all experiments are conducted using GPT-3.5-turbo [4, 26] to facilitate fair comparison and allow for future re-implementation.

### 4.2 Overall Performance Comparison

Table 1 presents the overall experimental results, we have the following observations: 1) The CORE methods demonstrate notable performance improvements when compared to both the GPT-3.5 variants and previous VQA models. Specifically, CORE methods outperform the GPT-3.5 counterparts ("A", "AE", "ALE") by 2.88%, 0.30%, and 0.94% respectively. This improvement is consistent across the different CORE enhancements applied to the GPT-3.5 model, indicating the effectiveness of the CORE framework in refining the

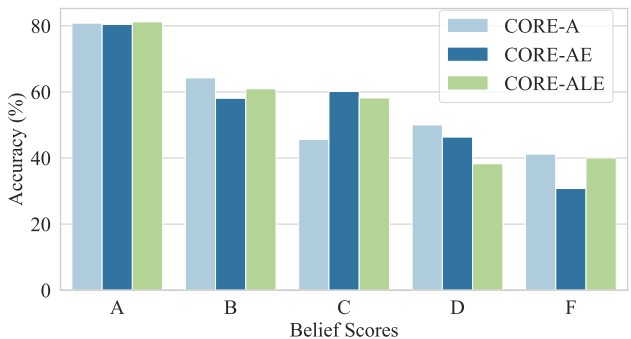

**Figure 4: Accuracy of different CORE models based on belief scores in initial QA responses on the ScienceQA dataset.**

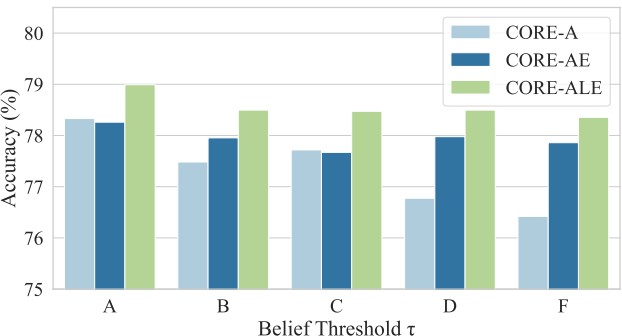

**Figure 5: The visualization of the performance of the three CORE methods across different belief threshold.**

capabilities of based models. 2) Of particular note is the CORE-A model, which shows the largest improvement in average performance. This suggests that applying the faithful reasoning of the CORE framework to the model with no reasoning output significantly enhances its ability to understand the knowledge required to solve the scientific questions, thereby improving overall performance. These gains highlight the CORE framework's potential to improve underlying model reasoning and comprehension in a multimodal context.

### 4.3 Effectiveness of Belief Modeling

In this section, we will delve into the key capabilities of the CORE approach in utilizing beliefs to accurately evaluate the truthfulness of initial answers. Figure 4 illustrates the accuracy of various belief scores within three models: CORE-A, CORE-AE, and CORE-ALE. From this data, we can draw two primary conclusions: 1) The highest performance is observed when the belief score is "A", exceeding 80% accuracy across all three models. This high level of accuracy indicates that when the models assign an "A" grade, reflecting high confidence in their responses, the predictions are very likely to be correct. Such strong performance at this confidence level confirms that the models are correctly identifying answers when they are most certain, validating the alignment between model confidence and answer correctness. 2) There is a clear trend of decreasing accuracy as the letter grades decline from "A" to "F" in each model. This gradient suggests that as the confidence level of the models decreases (reflected by lower letter grades), the probability of a VQA proposition as true knowledge also diminishes. This trend emphasizes the relationship between the model beliefs in their answers and the likelihood of those answers being correct, validating the reliability of using belief scores as an indicator of answer correctness.

### 4.4 Refinement w.r.t Different Belief Threshold

Given the effectiveness of belief modeling, it is essential to determine whether the refinement process should be applied universally to all initial predictions, including those with high belief scores. To this end, we aim to improve the model's initial outputs by refining responses that fall below a predetermined belief threshold $\tau$. Then we will generate a diverse answers $a'$ that align with the causal thought process $t_{v,q,a}$ and compare their beliefs with those of the initial output to select the one with higher belief as the final answer.

| Model | Avg Acc | Avg Steps | Avg Tokens |
|---|---|---|---|
| Standard | 45.0 | 1.00 | 52.29 |
| CoT | 44.1 | 1.55 | 149.5 |
| ReAct | 44.6 | 3.68 | 2408 |
| Standard + FR | 47.3 | 1.59 | 861.1 |
| CoT + FR | 47.6 | 2.18 | 1034 |
| ReAct + FR | 48.0 | 4.15 | 5646 |

**Table 2: Performance comparison of different reasoning models with and without the application of Faithful Reasoning (FR) on the HotpotQA dataset. The table reports average accuracy (Avg Acc), average number of reasoning steps (Avg Steps), and average number of tokens used (Avg Tokens) for Standard, Chain of Thought (CoT), and Reasoning and Acting (ReAct) models.**

Figure 5 illustrates the performance of three CORE methods across different belief thresholds (A, B, C, D, F), and there are two notable observations: 1) First, there is an observable increase in model performance as the belief threshold rises from "F" (lowest confidence) to "A" (highest confidence). This enhancement in performance highlights the refinement process's effectiveness in improving model accuracy, regardless of the starting confidence level. 2) When the threshold is set to "A", all three methods exhibit their highest performance levels. These findings indicate that refining responses, even when the initial belief is high, contributes positively to enhancing model performance. These findings support the strategy of refining model responses and estimating the corresponding beliefs, as shown in the unbiased estimation form 3, and then comparing their belief scores to determine the best final answer.

### 4.5 Effect on Different Prompting Techniques

To verify the effectiveness of our CORE framework in different prompting techniques, we further conduct ablation study of the faithful reasoning components (including Causal Reasoning, Belief Scoring, and Refinement) on the HotpotQA dataset, as shown in Table 2. We have the following observations: 1) The addition of Faithful Reasoning (FR) to all models results in a marked improvement in average accuracy. The most significant improvement is observed in the ReAct model, especially when integrated with FR. ReAct+FR achieves the highest accuracy, suggesting that our FR

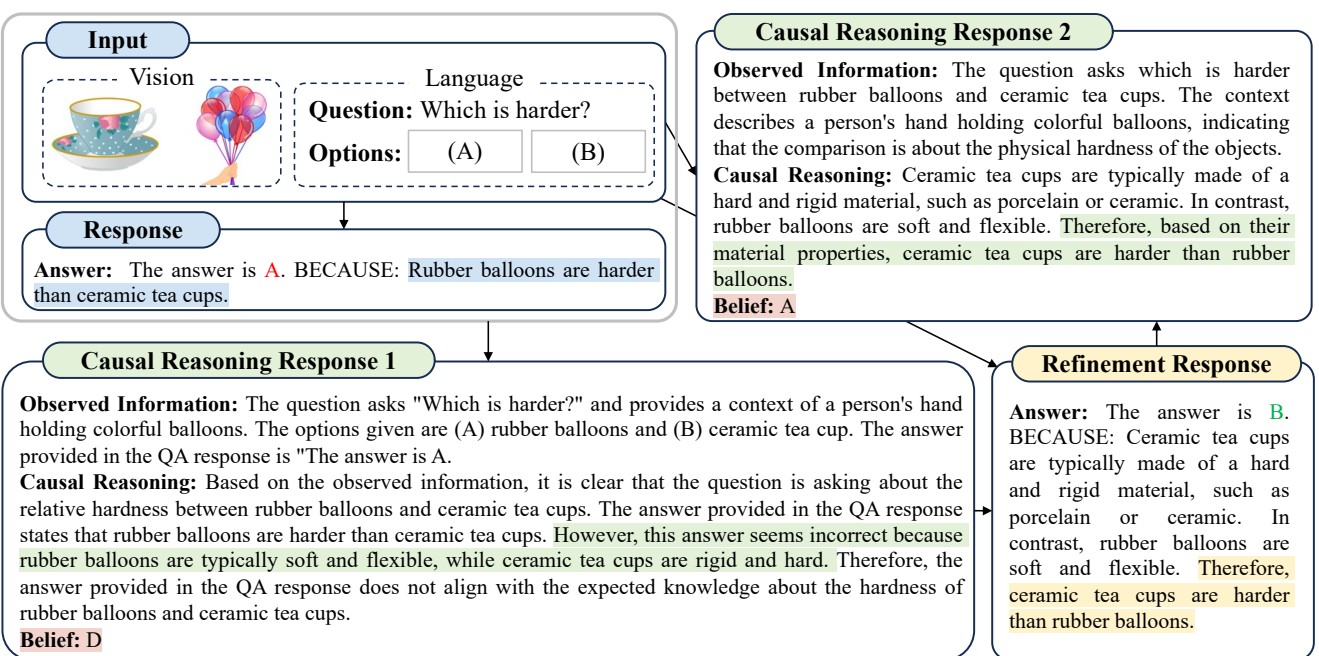

**Figure 6: The causal knowledge reasoning process of the CORE framework is illustrated by a case study, which demonstrates the transition from an initial hallucinatory output to the drawing of reasonable and accurate conclusions.**

method not only complements but also amplifies the effectiveness of complex reasoning approaches like ReAct. 2) An increase in the number of reasoning steps and tokens across the models indicates a more intricate reasoning process. While this complexity aids in achieving higher accuracy, it could also lead to longer inference times and potentially higher costs. These challenges present opportunities for future research aimed at improving the efficiency of FR implementations. This could involve exploring optimization techniques to streamline reasoning processes without compromising model performance. 3) The ReAct model, capable of executing tool interactions to acquire extrinsic knowledge, demonstrates enhanced performance with the addition of FR components. This improvement confirms the effectiveness of our model, especially by utilizing both intrinsic reasoning knowledge and extrinsic knowledge sources to improve the accuracy of reasoning.

## 4.6 Case Study

In Figure 6, a detailed case study illustrates the application of our CORE framework. The visual input for this instance consists two images: one of a ceramic cup and the other of a rubber balloon. The corresponding textual question posed is, "Which is harder?" with two choices presented.

Initially, the Question Answering module incorrectly selects Option A, rationalizing that "Rubber balloons are harder than ceramic cups." This incorrect response, misaligned with the actual scenario, illustrates a typical hallucination phenomenon in LLMs. At the same time, Causal Reasoning and Belief Scoring is utilized to evaluate this response. Upon analyzing the causal reasoning response 1, it assigns a D-level belief score, indicating a low level of confidence in the accuracy of the initial answer. Leveraging these insights, the Refinement module is engaged, utilizing both the initial response and

the Causal Reasoning Response 1 as inputs. It successfully revises the answer to Option B, providing a logically sound and accurate explanation. This revised answer is then re-evaluated through the Causal Reasoning, culminating in Causal Reasoning Response 2. This final response reaffirms the correctness of the revised answer and is awarded an A-level belief score, indicating high confidence in its accuracy.

This case study underscores the efficacy of the CORE framework in guiding and rectifying LLM-generated hallucinations. By sequentially applying the framework's components, the model effectively transitions from an erroneous to a correct and unbiased answer, guided by informed causal reasoning.

## 5 CONCLUSIONS AND FUTURE WORK

In conclusion, our CORE framework emerges as a unified solution for enhancing problem-solving and decision-making capabilities in LLMs, as evidenced by our comprehensive evaluations across two distinct datasets, ScienceQA and HotpotQA. At the heart of this framework are two key components: causal reasoning and belief scoring mechanisms. These components are essential for accurately assessing knowledge within the question answering scenarios.

While the success of framework is currently reflected through QA dataset accuracy, more direct evaluation of the causal reasoning and belief scoring components remains an area for further exploration. Future research should aim to develop methods to directly evaluate and optimize these modules. This direction will not only shed light on the inner workings of these mechanisms, but also facilitate the development of AI systems with efficient and transparent knowledge reasoning processes. Our work establishes a baseline for these developments, which are critical to evolving AI towards more nuanced, human-like reasoning capabilities.

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
