# OpenReview forum: "Causal-driven Large Language Models with Faithful Reasoning for Knowledge Question Answering"
_acmmm.org/ACMMM/2024/Conference — MM2024 Poster_

### Official Review · Reviewer_Bc8C · 2024-05-22

**Rating:** 4
**Confidence:** 2

**Summary:**

The author first constructs a causal graph that delineates the pathways between candidate knowledge and belief, and applies the do-calculus rule in the structural causal model to design an unbiased estimation framework(CORE), which is divided into four core modules: question answering, causal reasoning, belief scoring and refinement. Through the CORE framework, the occurrence of hallucinations can be effectively reduced, prompting the model to generate more accurate and reliable responses.

**Strengths:**

- Inspired by Goldman's causal theory, the author derived a set of causal knowledge reasoning framework suitable for large language models through formulas. This framework can establish a path from proposition to belief through causal connections, thereby improving the reliability and accuracy of LLM knowledge understanding.
- The author created an unbiased estimation framework, which contains four key modules: question answering, causal reasoning, belief scoring and refinement, which can allow LLM to obtain more faithful reasoning capabilities.

**Limitations:**

- At present, multi-modal large language models have achieved excellent results on ScienceQA. For example, the Avg index of the open source model LLaVA (13B) on ScienceQA has reached 90.92%[1], which is significantly different from the result in the author's paper (79.13%).  From this point of view, **how valuable is this research that needs further discussion?**
- From the paper, it is evident that the CORE framework has strong transferability. In the ScienceQA experiments, the authors converted image features into text. **Why not directly use image features and leverage powerful multimodal models such as LLaVA or GPT-4 Vision to conduct experiments?** This would better demonstrate the effectiveness of the CORE framework.
- According to line 4 of Algorithm 1 and section 3.2.4, **did the authors perform only one refinement?** If the confidence threshold is not met after the first refinement, will a second refinement be conducted? Could this iterative process potentially bring further improvements to the model's performance?
- In the causal reasoning ($P(t|v,q,a)$), the authors obtain causal thoughts $t$ based on visual-text features $v$, the question $q$, and the answer $a$. During the belief scoring ($P(b|v,q,a,t)$), they use visual features $v$, the question $q$, the answer $a$, and causal thoughts $t$ to obtain the belief score $b$. However, from the case in Figure 6 and Table 4 in the appendix, it seems that the authors used a single prompt input to simultaneously obtain Observed Information, Causal Reasoning, and Belief. **This appears to be inconsistent with the formula representations.**
- In Table 1, the GPT-3.5-ALE achieved the best performance in the LAN column, which was incorrectly labeled.

[1] https://scienceqa.github.io/leaderboard.html

**Suitability:**

2

---

### Official Review · Reviewer_d6zt · 2024-05-23

**Rating:** 6
**Confidence:** 3

**Summary:**

The paper presents a novel framework called CORE (Causal knOwledge REasoning) for knowledge question answering in large language models (LLMs). The framework combines the principles of causal theory to enhance the reliability and accuracy of knowledge understanding in LLMs. The paper proposes four essential components of the CORE framework: question answering, causal reasoning, belief scoring, and refinement. Extensive experiments on the ScienceQA and HotpotQA datasets demonstrate the effectiveness and rationality of the CORE framework.

**Strengths:**

1) The paper introduces a novel framework that integrates causal reasoning into large language models for knowledge question answering, which successfully improves the trustfulness and performance of existing LLMs
2) It is interesting to see the theory behind the designing of the proposed pipeline. The connections between causal reasoning and the proposed framework look reasonable to me, which also can be used to explain the motivation of other similar LLM reasoning piplines
3) The paper provides a well-structured and comprehensive explanation of the CORE framework. Experiment results demonstrate the effectiveness of the proposed method.

**Limitations:**

1) The proposed framework is not novel. There are some similar works following the reasoning, support generation, evaluation, and refinement pipeline. [1] However, I think the novelty of this paper is actually to provide a potential theory explanation for this kind of pipeline.

2) The details of the methods can be further explained. For example, how Eq 3 is connected to the proposed framework can be further discussed. How many refinement loops are conducted? Can the refinement loops improve the confidence?

3) There are some potential typos that should be fixed: In Alg. 1 Line 4, should the belief score lower than the threshold? In the experiment, the author use GPT-3.5 as the multi-modal model. However, to the best of my knowledge, GPT-3.5 cannot take images as input. It is better for authors to present a link to the model they used in the implementation setting.


[1] SuRe: Summarizing Retrievals using Answer Candidates for Open-domain QA of LLMs, ICLR 2024

**Suitability:**

3

---

### Official Review · Reviewer_fPnZ · 2024-05-25

**Rating:** 2
**Confidence:** 4

**Summary:**

This paper proposes a novel framework that incorporates Causal Inference to the generation of LLM, which can enhance the knowledge and understanding ability of LLM. Drawing inspiration from Causal Inference, this paper adopts `do` operations on V Q and A to derive an unbiased belief estimation. This study conducted experiments on ScienceQA and HotpotQA datasets to demonstrate the effectiveness of their methods.

**Strengths:**

- Each part of their design is intrinsically related and well-illustrated, making the framework easy to follow.
- The vivid and well-illustrated case study makes this paper more interesting and easy to understand.
- It dives into the critical issue of trustworthiness and robustness in LLMs.

**Limitations:**

- The major weakness pertains to the causal graph proposed by the paper.
- The edges from {U->VQ} in the causal graph contradict the derivation of the equations.
- The unobserved confounder U lacks a more detailed explanation or a more specific division.
- In the Introduction, the introduction and separation of intrinsic and extrinsic knowledge are confusing, since the Methods include few, if any, specific operations corresponding to two kinds of knowledge.
- In Related Work, an explanation of causal inference is missing, making it difficult for readers to understand quickly. Adding some background information about causal inference techniques [1][2] is recommended.
- In Experiments, the baseline models used are outdated. It is advisable to evaluate more recent methodologies, such as MM-CoT[3], which has demonstrated an average accuracy of 90.45 on the ScienceQA.

**Suitability:**

3

---

### Official Review · Reviewer_gf7y · 2024-05-27

**Rating:** 4
**Confidence:** 3

**Summary:**

This paper studies LLMs for KQA problems by introducing an unbiased estimation framework based on Goldman’s causal theory of knowledge. The proposed CORE framework consists of four essential components: question answering, causal reasoning, belief scoring, and refinement. Experiments are conducted on ScienceQA and HotpotQA.

**Strengths:**

The proposed method introduces a new causal theory grounding to solve LLM KQA problems. The paper is well-organized and easy to follow.

**Limitations:**

Q1. In Section 3.1, the authors demonstrate how the causal graph is built based on Peal’s structural causal model. What remains confusing to me is how Goldman’s causal theory contributes to this causal modeling and the CORE framework.

Q2. What LLM do you use in the causal reasoning component?

Q3. According to Lines 485-488, could you showcase the chain-of-thought detailed prompts and the generated response with introspection and causal awareness?

Q4. The framework is claimed to be estimated “within the context of the VQA task”, but the authors choose HotpotQA as the second evaluated dataset. Could you please explain the motivation for dataset selection? Moreover, I recommend the authors should consider a consistent evaluation of challenging VQA tasks such as MMMU [1].

Q5. In Table 1, the authors should compare CORE with the CoT methods in the ScienceQA leaderboard [2].

[1] Yue X, Ni Y, Zhang K, et al. Mmmu: A massive multi-discipline multimodal understanding and reasoning benchmark for expert agi[J]. arXiv preprint arXiv:2311.16502, 2023.

[2] https://scienceqa.github.io/leaderboard.html

**Suitability:**

2

---

### Meta-Review · Area_Chair_GWsm · 2024-07-08

**Recommendation:** Accept (Poster)
**Confidence:** 4

**Metareview:**

The paper introduces a novel framework called CORE, which integrates causal reasoning into large language models for knowledge question answering. While it is well-organized and addresses trustworthiness and robustness issues in LLMs, reviewers raised concerns about the causal graph and unobserved confounders. The ratings varied from weak reject to accept, and recommendations include addressing these concerns and providing more detailed explanations of methods and dataset choices. The final recommendation is acceptance. Authors are encouraged to address the concerns raised by the reviewers in the final version.